# Hand Osteoarthritis in the Elderly: The Prevalence of Articular Cartilage Defects in Radiographically Normal and Affected Joints

**DOI:** 10.3390/diagnostics15212669

**Published:** 2025-10-22

**Authors:** Reiji Nishimura, Tohru Hashimoto, Takeshi Fukuda, Tohru Yano, Kazuhiro Maeda, Masataka Okabe, Takeshi Miyawaki

**Affiliations:** 1Department of Plastic and Reconstructive Surgery, The Jikei University School of Medicine, Tokyo 105-8461, Japan; caritakm@jikei.ac.jp; 2Department of Anatomy, The Jikei University School of Medicine, Tokyo 105-8461, Japan; tohruyano@jikei.ac.jp (T.Y.); maokabe@jikei.ac.jp (M.O.); 3Department of Radiology, The Jikei University School of Medicine, Tokyo 105-8471, Japan; takenet616@gmail.com; 4Department of Orthopedics, The Jikei University School of Medicine, Tokyo 105-8461, Japan; maeda@jikei.ac.jp

**Keywords:** osteoarthritis, hand, cartilage, radiography, diagnosis

## Abstract

**Background**: Hand osteoarthritis (OA) is a highly prevalent disease that significantly impairs quality of life among many patients. The direct evaluation of cartilage defects associated with OA in vivo is challenging, and indirect assessments using X-ray images are commonly employed. The aim of this study was to evaluate the relationship between X-ray images of finger joints and cartilage defects. **Methods**: This study included 42 hands from cadavers that were fixed with alcohol and formalin. After X-ray posteroanterior images of all the finger joints were taken, the extent of the cartilage defects was observed macroscopically. On X-ray images, OA was defined as a modified Kellgren–Lawrence scale score of 2 or higher. Histological examinations were performed on several joints with cartilage defects to confirm whether the macroscopic cartilage defects corresponded to the histological cartilage defects. **Results**: A total of 588 joints were evaluated. On X-ray images, OA was observed in 20.2% of the joints, and cartilage defects were present in 45.1%. Among joints with cartilage defects, the prevalence of joints without radiographic OA was 55.1%. On the other hand, 31.1% of joints without radiographic OA had cartilage defects. Cartilage defects were identified in all joints with radiographic OA. **Conclusions**: The use of X-ray images to evaluate OA is beneficial; however, when interpreting radiographic OA, it is important to note that early or partial OA may not be detectable. Additionally, when OA findings are present on X-ray images, cartilage defects are always present.

## 1. Introduction

Osteoarthritis (OA) is a chronic disease with a high global prevalence, and the pain and functional impairment caused by this disease can significantly impair daily activities and work [1,2]. There are few effective preventive or therapeutic options for OA; therefore, this disease continues to impose a significant societal burden [1,2]. The hands are among the body parts with the highest prevalence of OA, along with the knees and hips [3,4]. Pain from hand OA can significantly impair daily activities, and joint deformities caused by OA may lead to cosmetic concerns [5].

OA is a pathological condition of the joint that is characterized by localized cartilage loss in synovial joints, accompanied by bone enlargement and joint capsule thickening [6]. Cartilage loss (eburnation) associated with OA can be confirmed by completely exposing the joint surface of the sample [7,8]. The Mankin grading and the OARSI osteoarthritis cartilage histopathology assessment system was reported as histological indicators for evaluating the progression of OA [9,10]. However, directly evaluating intra-articular conditions in vivo is challenging; therefore, OA is typically diagnosed indirectly using images and symptoms [6]. X-ray images, in particular, are simple and minimally invasive; thus, they are widely used in clinical and epidemiological studies [11,12,13]. On the other hand, reports investigating the relationship between findings on X-ray images and actual intra-articular findings such as cartilage defects are limited. Sunk et al. compared Kellgren and Lawrence (K–L) scores and Mankin scores and reported the usefulness of X-ray-based OA assessment [4]. The aim of this study was to investigate the relationship between findings on X-ray images and joint cartilage defects in OA of the digits.

## 2. Materials and Methods

This study was conducted using 21 cadavers (42 hands) from Japanese individuals fixed with ethanol and formalin, and stored at room temperature inside a steel container. There were 7 males and 14 females, with an age range of 69–105 years and an average age of 85.5 years. Cadavers with a history of rheumatoid arthritis were excluded. Ethical approval was obtained from the Institutional Review Board prior to the initiation of this study (approval number: 32-425, approval date: 4 July 2022). In this study, written informed consent has been obtained from the all donors of cadavers to be included in this study and publish the result. All procedures in this study were conducted in accordance with the Declaration of Helsinki.

Posteroanterior X-ray images of all the finger joints were taken. To obtain accurate X-ray images of all the finger joints, multiple images were taken of the same hand. The X-ray images were evaluated using the modified K L classification. Each joint was graded on a scale ranging from 0–4: 0 = no OA (no osteophyte or joint space narrowing), 1 = questionable osteophyte or joint space narrowing, 2 = small osteophyte(s) or mild joint space narrowing, 3 = moderate osteophyte(s) or joint space narrowing, and 4 = large osteophyte(s) or joint space narrowing [14]. Joints with a score of 2 or higher were defined as having radiographic OA. Blinded readings of all joints were performed by a musculoskeletal radiologist (TF). The same joints were blinded and read by a hand surgery specialist (RN). The interrater agreement was determined by calculating the kappa value (0.842). The radiologist’s evaluation was used for analysis, and the hand surgeon’s evaluation was used only to calculate the inter-rater agreement.

Subsequently, the articular cartilage was observed and evaluated. To observe the articular cartilage defects, the skin on the dorsal side of the hand was removed, and the dorsal joint capsule and both lateral collateral ligaments were dissected at each joint to fully expose the articular surface (Figure 1). The condition of each joint surface was visually inspected, and the distribution of the articular cartilage defects was recorded. All the anatomical dissections and observations were performed by a hand surgeon (RN). The joint surface where cartilage is preserved and the joint surface where the cortical bone is exposed due to cartilage loss can be easily distinguished by differences in color and texture. To confirm the reliability of the macroscopic evaluation, a histological evaluation was performed on 12 joints randomly selected from those where the cartilage defects were identified. Joint samples were refixed with 10% formalin solution, dehydrated and degreased with 100% ethanol, decalcified with Plank-Rychlo solution, neutralized with 5% sodium sulfate, dehydrated with 70–100% ethanol after washing with pure water, embedded in paraffin using an embedding apparatus (SAKURA SEIKI, Tissue-Tek VIP-5Jr, Nagano, Tokyo and Tokyo, Japan), sectioned into 5 μm-thick slices, stained with toluidine blue, and examined under a microscope.

The results of the joint evaluation were used to summarize the rate of radiographic OA and articular cartilage defects, and then the rate of joints without radiographic OA in joints with cartilage defects (false negative rate [FNR] of radiographic OA), the rate of articular cartilage defects in joints without radiographic OA (1-negative predictive value [NPV] of radiographic OA), and the rate of cartilage defects in joints diagnosed with radiographic OA (positive predictive value [PPV] of radiographic OA) were determined. These figures were then used to analyze the relationship between X-ray images of finger joints and cartilage defects. All statistical analyses were performed using the chi-square test, which is an appropriate method for testing the difference in proportions between two groups consisting of non-paired nominal scale data with sufficient expected frequencies. For the analysis, *p*-value (*p*), odds ratio (OR) and 95% confidence interval (CI) were calculated. The significance level for *p* was set at 0.05.

All figures were saved in the LZW-compressed high-resolution TIFF file format with a resolution of 350 dpi.

## 3. Results

### 3.1. Histological Confirmation of Cartilage Defects

Histological examination using toluidine blue staining of joints with macroscopically visible cartilage defects revealed clear distinctions between areas where the cartilage remained intact and areas where the cartilage was completely absent (eburnation) (Figure 2). In areas where cartilage was deficient, subchondral bone thickening was observed. In all 12 joints examined histologically, the macroscopic cartilage defects corresponded to the histological cartilage defects.

### 3.2. Incidence of Radiographic OA and Cartilage Defects

Radiographic OA (KL grade ≥ 2) was detected in 20.2% of all digit joints, and in 42.4% of the IP and DIP joints, 15.5% of the PIP joints, and 1.9% of the MP joints (Figure 3, Table 1). Radiographic OA was present in 10.2% of joints among males and 25.3% among females, with a higher prevalence among females (*p* < 0.0001, OR = 2.97, 95% CI: 1.78–4.98). In the IP and DIP joints, the prevalence of radiographic OA was higher among females than among males (*p* < 0.0001, OR = 4.11, 95% CI: 2.12–7.96), and a similar tendency was observed in the PIP joints (*p* = 0.0347, OR = 3.18, 95% CI: 1.04–9.73).

Cartilage defects were observed in 45.1% of all the finger joints, 73.8% of the IP and DIP joints, 38.7% of the PIP joints, and 21.4% of the MP joints. Cartilage defects were present in 35.2% of joints among males and 50.0% among females, with a higher prevalence among females (*p* = 0.0007, OR = 1.84, 95% CI: 1.29–2.62). In the IP and DIP joints, the prevalence of cartilage defects was higher among females than among males (*p* = 0.0039, OR = 2.51, 95% CI: 1.33–4.74), and a similar tendency was also observed in the MP joints (*p* = 0.0125, OR = 2.78, 95% CI: 1.22–6.36).

The sensitivity of X-ray images for cartilage defects was 44.9% (95% CI: 38.9–50.9%), specificity was 100% (95% CI: 99.8–100%), PPV was 100% (95% CI: 99.5–100%), and NPV was 68.9% (95% CI: 64.7–73.1%).

Among the joints with cartilage defects, the rate of joints without radiographic OA (FNR) was 55.1%; 42.6% of these joints were IP/DIP joints, 60.0% were PIP joints, and 91.1% were MP joints (Table 2). The FNR among the MP joints was higher than those among the IP/DIP joints (*p* < 0.0001, OR = 13.8, 95% CI: 4.71–40.50) and PIP joints (*p* = 0.0003, OR = 6.83, 95% CI: 2.18–21.37).

Localized cartilage defects that were not diagnosed as OA via X-ray images were more common in the MP joint than in the other joints (Figure 4). The FNR among males was higher than those among females in the IP/DIP joints (*p* = 0.0004, OR = 3.64, 95% CI: 1.74–7.61). Among joints without radiographic OA (KL grade ≤ 1), the prevalence of joints with cartilage defects (1-NPV) was 31.1%. The 1-NPV was 54.5% among IP/DIP joints, 27.5% among PIP joints, and 19.9% among MP joints (Table 3). Among the MP joints, a trend toward a higher 1-NPV among females than among males was observed (*p* = 0.0128, OR = 2.92, 95% CI: 1.22–7.00). All joints diagnosed with OA via X-ray images presented with cartilage defects, resulting in a PPV of 100%.

## 4. Discussion

### 4.1. Evaluation Methods for OA

OA is a multifactorial and heterogeneous disease [15,16,17,18,19]. The prevalence of OA varies based on the definition and the affected site [4]. OA can be defined as a condition characterized by localized loss of articular cartilage in synovial joints, accompanied by bone hypertrophy (osteophytes + subchondral bone sclerosis) and synovial membrane thickening [6]. A more comprehensive definition describes OA as a disorder involving movable joints characterized by cell stress and extracellular matrix degradation initiated by micro- and macro-injury that activates maladaptive repair responses, including proinflammatory pathways of innate immunity [20]. In OA studies, three definitions have been commonly used: “radiographic OA”, “symptomatic OA”, and “self-reported OA” [6,21,22]. Among these, the most widely used definition is X-ray images [6]. While X-ray images can only indirectly assess structural changes in joints associated with OA, they are inexpensive, quick, and easy to use for evaluating lesions [4]. Radiographic OA and symptomatic OA are not necessarily correlated [4,6,23].

In previous cohort studies, the prevalence of radiographic OA ranged from 36% to 91%, whereas the prevalence of symptomatic OA ranged from 3% to 17%, indicating that radiographic OA without symptoms is common [5,12,15,24]. Large-scale cohort studies using X-ray images have revealed the following characteristics of the distribution of hand OA: hand OA is more common among men until middle age and more common among females thereafter [11,15,25]. There is a strong tendency for accumulation in the same row (same hand, same joint), with a higher prevalence rate among DIP, thumb IP, and thumb CM joints [11,26]. There is a tendency for accumulation within the same ray (same ray of digits) [11,26]. There is a tendency for symmetrical accumulation (the same joints on both sides) [11,26]. In this study, overall radiographic OA was more common among females than among males, and a tendency toward accumulation in the DIP joints along the row was observed.

If it were possible to directly observe the intra-articular structures containing cartilage, it would be possible to accurately assess the changes in joints associated with OA. However, direct observation of the intra-articular space is too invasive for clinical diagnosis or epidemiological studies. Therefore, opportunities to obtain data directly evaluating articular cartilage are limited to studies using cadavers or specimens excised during surgery. Furthermore, reports comparing such histological changes with evaluations on X ray images are very limited. Mankin et al. proposed a histological grading system for OA based on four criteria, namely, structure, cells, safranin-O staining, and tide marks; this grading system has become widely used [9]. However, the Mankin system fails to reflect changes in the early stages of OA, prompting Plitzker et al. to propose a more standardized grading system based on changes in articular cartilage [10]. Pellegrini et al. evaluated the articular surfaces of thumb CMC joints harvested during surgery using macroscopic examination and photography and recorded the extent of cartilage defects [8]. In this report, cases with more advanced OA stages on X-ray images presented a wider range of cartilage defects, particularly on the metacarpal base side. Sunk et al. performed histological evaluation (Mankin score) and evaluation on X ray images (KL score) of DIP and PIP joints obtained from cadavers and reported that the two were correlated [14]. However, the histological OA diagnostic criteria in this report used a cut-off value of a Mankin score ≥ 5 and included stages prior to cartilage defects in the histological OA diagnosis. This study clarifies the FNR in detecting cartilage defects by X-ray images and provides new insights into the relationship between joint evaluation using X-ray images and changes in cartilage within the joint.

### 4.2. Differences in the Timing of Cartilage Defect and Radiographic OA Development

In this study, 55.1% of the joints with cartilage defects did not have radiographic OA findings (FNR), and 31.1% of the joints not diagnosed with radiographic OA had cartilage defects (1-NPV). On the other hand, 100% of the joints diagnosed with OA via X-ray images had cartilage defects (PPV). This may be because the phases in which cartilage defects occur and those in which radiographic OA is detected differ in the progression of OA. In the early stages of the OA process, the cartilage in areas of the joint surface that bears the highest mechanical load is lost locally, whereas the cartilage structure in other areas remains intact [10]. As OA progresses, changes spread throughout the entire joint surface. Direct observation of the joint surface can reveal localized cartilage defects in the early stages of OA. On the other hand, significant narrowing of the joint space, subchondral bone thickening, and osteophyte formation—changes that are identifiable on X-ray images—become evident in more advanced stages of OA. Additionally, finger joints are much smaller than knee or hip joints, resulting in a relatively lower resolution of X-ray images, which makes it difficult to identify early changes associated with OA. Furthermore, since cartilage defects were identified in all joints diagnosed with OA via X-ray images, it can be concluded that cartilage defects are almost always present when OA findings are observed on X-ray images.

### 4.3. OA of the MP Joint

In this study, the percentage of joints without radiographic OA in joints with a cartilage defect (FNR) was higher among MP joints than among the IP/DIP joints and PIP joints. This result suggests that cartilage defects in the MP joint progress to the point of showing OA findings on X-ray images less frequently than those in the IP, DIP, and PIP joints. Unlike IP joints, which perform flexion-extension movements within the same plane, MP joints perform three-dimensional movements that include internal and external rotation, resulting in a more three-dimensional joint surface shape. This difference in joint shape and the associated differences in intra-articular load distribution may make it difficult to detect OAs via X-ray images, which is a two-dimensional projection.

While females have a higher prevalence of radiographic OA in all finger joints, previous studies have shown that there is no gender difference in the MP joint [10] or that males have a higher prevalence [13,15]. This finding is speculated to be due to the MP joint being more susceptible to excessive loading and trauma than the distal PIP and DIP joints [27]. In the present study, no gender difference was detected in the prevalence of radiographic OA in the MP joints (*p* = 0.7211). The prevalence of cartilage defects in the MP joints and the proportion of cartilage defects not diagnosed as radiographic OA (1-NPV) were both higher among females. This phenomenon suggests that while early stage OA (OA with only partial cartilage defects) is more likely to occur in females, the prevalence of progression to a stage exceeding the X-ray detection threshold is higher in males. If this is the case, excessive physical load on the MP joint may influence the progression of OA rather than its onset.

### 4.4. Clinical Relevance

OA assessment using X-ray images is very simple and standardized; therefore, even if localized cartilage defects cannot be detected, their clinical usefulness remains unchanged. On the other hand, when X-ray images are interpreted for suspected OA, it is important not to exclude the possibility that joints not diagnosed with OA on X-ray images may have partial cartilage defects, and that these partial cartilage defects may cause symptoms such as pain and swelling.

Diagnosing early stage OA before it progresses to the established-stage detectable by X-ray images is crucial for implementing treatments that slow or reverse OA progression [28]. Ultrasonography and magnetic resonance imaging have been reported as strong candidates for non-radiographic imaging OA evaluation methods [29,30,31]. While detecting partial cartilage defects with these imaging modalities remains challenging at present, advances in high spatial resolution imaging techniques are expected to facilitate the detection of early stage OA [32]. Combining these modalities with X-ray images is anticipated to provide complementary value. Furthermore, integrating artificial intelligence assistance holds potential for enhancing the accuracy of OA diagnosis [33].

### 4.5. Limitations

This study has several limitations. The number of cases was limited, making accurate evaluation of the distribution of cartilage defects and radiographic OA difficult. However, the prevalence and distribution of radiographic OA did not differ significantly from those reported in previous large cohort studies. Due to the lack of information on trauma history, it was not possible to distinguish between spontaneous OA and trauma-related OA. Additionally, due to the absence of information on dominant hand, occupation, and symptoms, subgroup analyses based on these variables could not be performed. Furthermore, as this study utilized donated cadavers, the study population was limited to elderly individuals. This study was limited to Japanese elderly individuals without information regarding symptoms. Therefore, the results of this study do not directly reflect the conditions of groups with different backgrounds, such as populations experiencing symptoms like pain, groups of different age ranges, or other ethnic groups.

## 5. Conclusions

Hand OA is a highly prevalent disease, making its diagnosis and treatment crucial. X-ray images are widely used for clinical OA diagnosis and lesion distribution investigation because they are simple and minimally invasive. However, the information on joint surfaces obtained from X-ray images is partial and indirect. The results of this study suggest that the diagnosis of OA on X-ray images often fails to detect cartilage defects on the articular surface. While X-ray images remain useful for diagnosing OA, caution is required when interpreting negative findings. For patients suspected of having hand OA without radiographic evidence, diagnosing OA based solely on symptoms or using other modalities should be considered.

## Figures and Tables

**Figure 1 diagnostics-15-02669-f001:**
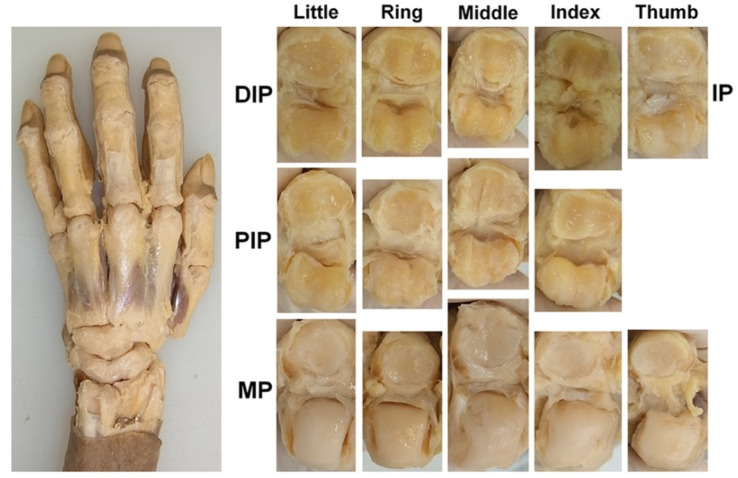
Exposure and observation of the joint cartilage; the DIP, PIP, MP, and IP joints of the digits were dissected, and all the joint surfaces were observed directly.

**Figure 2 diagnostics-15-02669-f002:**
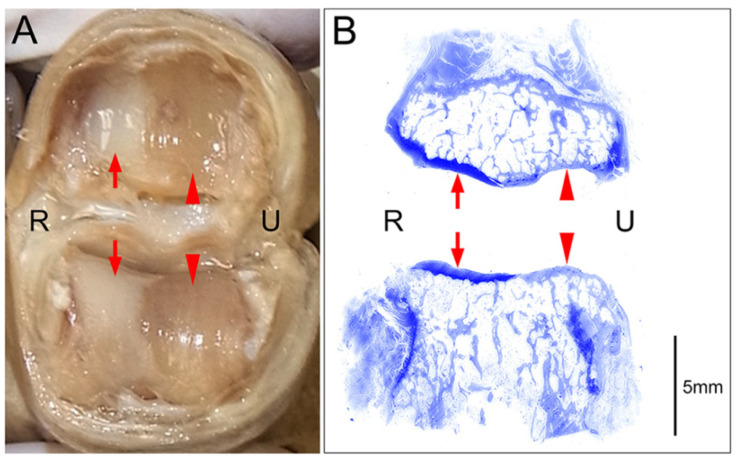
An example of a cartilage defect at the joint surface (PIP joint of the right middle finger); R: radial side, U: ulnar side. (**A**) Macroscopic image of the joint surface showing preserved cartilage layers on the radial side (arrows) and complete cartilage defects (eburnation) on the ulnar side (arrowheads). (**B**) Coronal tissue sections of the specimen presented in A stained with toluidine blue after decalcification are shown. Histological examination of these sections revealed maintained cartilage layers that were stained dark with toluidine blue on the radial side (arrows) and cartilage defects with consequent thickening of the subchondral bones that were stained pale with toluidine blue on the ulnar side (arrowheads).

**Figure 3 diagnostics-15-02669-f003:**
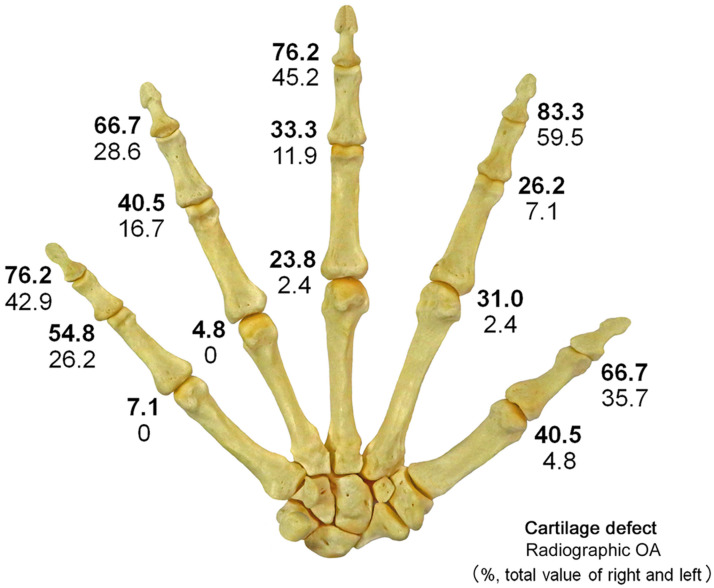
Distribution of cartilage defects and radiographic OA; The figure shows the prevalence (%) of cartilage defects and radiographic OA caused by joints combined for both sides. The upper numbers indicate cartilage defects (bold), and the lower numbers indicate radiographic OA. Cartilage defects and radiographic OA are most common in DIP joints and relatively less common in MP joints. In all joints, the prevalence of cartilage defects was higher than that of radiographic OA.

**Figure 4 diagnostics-15-02669-f004:**
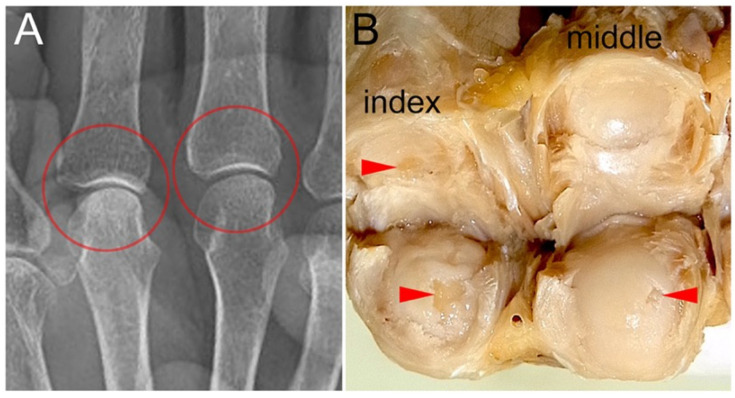
MP joint surfaces with cartilage defects and their X-ray images: (**A**) A X-ray image of the gross sample presented in B, taken prior to dissection, is shown. No changes suggestive of OA are observed in the MP joints of the index and middle fingers (circled areas). (**B**) Macro-anatomical examination revealed cartilage defects in the MP joints of the index and middle fingers (arrowheads).

**Table 1 diagnostics-15-02669-t001:** Prevalence of radiographic OA and cartilage defects. This table shows the percentages of radiographic OA and cartilage defects in the thumb IP, DIP, PIP, and MP joints for each gender. Both radiographic OA and cartilage defects were more common among females overall.

	Radiographic OA (%)			Cartilage Defects (%)		
	Both Genders (*N* = 42/Each Finger)	Male (*N* = 14/Each Finger)	Female (*N* = 28/Each Finger)	Both Genders (*N* = 42/Each Finger)	Male (*N* = 14/Each Finger)	Female (*N* = 28/Each Finger)
IP·DIP						
Thumb	35.7	0	53.6	66.7	50.0	75.0
Index	59.5	50.0	64.3	83.3	71.4	89.3
Middle	45.2	21.4	57.1	76.2	64.3	82.1
Ring	28.6	0	42.9	66.7	57.1	71.4
Little	42.9	35.7	46.4	76.2	64.3	82.1
All fingers	42.4	21.4	52.9	73.8	61.4	80.0
PIP						
Index	7.1	7.1	7.1	26.2	28.6	25.0
Middle	11.9	14.3	10.7	33.3	35.7	32.1
Ring	16.7	0	25.0	40.5	14.3	53.6
Little	26.2	7.1	35.7	54.8	50.0	57.1
All fingers	15.5	7.1	19.6	38.7	32.1	42.0
MP						
Thumb	4.8	0	7.1	40.5	35.7	42.9
Index	2.4	0	3.6	31.0	14.3	39.3
Middle	2.4	7.1	0	23.8	7.1	32.1
Ring	0	0	0	4.8	0	7.1
Little	0	0	0	7.1	0	10.7
All fingers	1.9	1.4	2.1	21.4	11.4	26.4
Total						
	20.2	10.2	25.3	45.1	35.2	50.0

**Table 2 diagnostics-15-02669-t002:** FNR of radiographic OA for cartilage defects. This table shows the FNR for each joint, broken down by gender. This FNR represents the proportion of joints with cartilage defects that are not detected as OA on X-ray images.

	Number of Joints with Cartilage Defects on Normal X-Rays/Number of Joints with Cartilage Defects			FNR (%)		
	Both Genders	Male	Female	Both Genders	Male	Female
IP·DIP						
Thumb	13/28	7/7	6/21	46.4	100	28.6
Index	10/35	3/10	7/25	28.6	30.0	28.0
Middle	13/32	6/9	7/23	40.6	66.7	30.4
Ring	16/28	8/8	8/20	57.1	100	40.0
Little	14/32	4/9	10/23	43.8	44.4	43.5
All fingers	66/155	28/43	38/112	42.6	65.1	33.9
PIP						
Index	8/11	3/4	5/7	72.7	75.0	71.4
Middle	9/14	3/5	6/9	64.3	60.0	66.7
Ring	10/17	2/2	8/15	58.8	100	53.3
Little	12/23	6/7	6/16	52.2	85.7	37.5
All fingers	39/65	14/18	25/47	60.0	77.8	53.2
MP						
Thumb	15/17	5/5	10/12	88.2	100	83.3
Index	12/13	2/2	10/11	92.3	100	90.9
Middle	9/10	0/1	9/9	90.0	0	100
Ring	2/2	0/0	2/2	100	-	100
Little	3/3	0/0	3/3	100	-	100
All fingers	41/45	7/8	34/37	91.1	87.5	91.9
Total						
	146/265	49/69	97/196	55.1	71.0	49.5

**Table 3 diagnostics-15-02669-t003:** 1-NPV of radiographic OA. This table shows the 1-NPV for each joint, broken down by gender. This 1-NPV represents the proportion of joints where cartilage is present among those that are not diagnosed as OA on X-ray images.

	Number of Joints with Cartilage Defects on Normal X-Rays/Number of Normal Joints on X-Rays			1-NPV (%)		
	Both Genders	Male	Female	Both Genders	Male	Female
IP·DIP						
Thumb	13/27	7/14	6/13	48.1	50.0	46.2
Index	10/17	3/7	7/10	58.8	42.9	70.0
Middle	13/23	6/11	7/12	56.5	54.5	58.3
Ring	16/30	8/14	8/16	53.3	57.1	50.0
Little	14/24	4/9	10/15	58.3	44.4	66.7
All fingers	66/121	28/55	38/66	54.5	50.9	57.6
PIP						
Index	8/39	3/13	5/26	20.5	23.1	19.2
Middle	9/37	3/12	6/25	24.3	25.0	24.0
Ring	10/35	2/14	8/21	28.6	14.3	38.1
Little	12/31	6/13	6/18	38.7	46.2	33.3
All fingers	39/142	14/52	25/90	27.5	26.9	27.8
MP						
Thumb	15/40	5/14	10/26	37.5	35.7	38.5
Index	12/41	2/14	10/27	29.3	14.3	37.0
Middle	9/41	0/13	9/28	22.0	0	32.1
Ring	2/42	0/14	2/28	4.8	0	7.1
Little	3/42	0/14	3/28	7.1	0	10.7
All fingers	41/206	7/69	34/137	19.9	10.1	24.8
Total						
	146/469	49/176	97/293	31.1	27.8	33.1

## Data Availability

The data presented in this study are available on request from the corresponding author due to privacy and ethical reasons.

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
