# Peer review of "Hand Osteoarthritis in the Elderly: The Prevalence of Articular Cartilage Defects in Radiographically Normal and Affected Joints"

_diagnostics, 2025, doi:10.3390/diagnostics15212669_

Round 1
Reviewer 1 Report
Comments and Suggestions for Authors
This paper studies an important diagnostic limitation in osteoarthritis (OA): the gap between radiographic findings and actual cartilage damage. While hand OA is highly prevalent and debilitating, most clinical and epidemiological data rely on imaging methods as plain radiography. The novelty of this work lies in its direct comparison between radiographic scoring (modified Kellgren–Lawrence scale) and macroscopic cartilage evaluation in cadaveric hand joints, confirmed by histology.
The article provides high-quality anatomical and diagnostic data, with important methodology and adequate sample size for descriptive conclusions (588 joints).
The limitations of your study are :
-
The study is restricted to elderly Japanese cadavers, which may not fully represent the demographic or clinical spectrum of OA.
-
No correlation with symptomatic OA (pain, function), which is clinically more relevant than radiographic or histological findings alone.
-
Imaging limited to plain radiographs; comparison with more advanced modalities (MRI, CT)
Despite these limitations, the information quality is consistent and helpfully for acquiring knowledge about diagnostic reliability of radiography in hand OA.
Your paper is a high-quality, well-structured cadaveric study with interesting insights into the limitations of radiography for hand OA diagnosis.
Author Response
Comments1: The limitations of your study are :
The study is restricted to elderly Japanese cadavers, which may not fully represent the demographic or clinical spectrum of OA.
No correlation with symptomatic OA (pain, function), which is clinically more relevant than radiographic or histological findings alone.
Imaging limited to plain radiographs; comparison with more advanced modalities (MRI, CT)
Despite these limitations, the information quality is consistent and helpfully for acquiring knowledge about diagnostic reliability of radiography in hand OA.
Response1: Thank you for your advice. We completely agree with your opinion regarding our limitations. In line with the advice of other reviewers, I added the following to the limitations: (This study is limited to Japanese elderly individuals without information regarding symptoms) " Therefore, the results of this study do not directly reflect the conditions of groups with different backgrounds, such as populations experiencing symptoms like pain, groups of different age ranges, or other ethnic groups."
Reviewer 2 Report
Comments and Suggestions for Authors
This manuscript dicuss an important clinical problem: the burden of osteoarthritis (OA), specifically hand OA, and the limitations of current diagnostic modalities. This topic is clinically relevant and of potential interest to readers.
Sentence length can be shortened.
Avoid redundancy.
Introduction
Please restructure following a logical flow:
Burden and clinical importance of OA
Focus on hand OA
Pathological understanding and histological assessments
Limitations of current diagnostic tools
Study aim.
Materials and Methods
There is no mention of ethical approval or institutional oversight for cadaveric research.
Expand details on imaging parameters, fixation/storage, and histology methods for reproducibility.
Provide justification for statistical choices and, if feasible, report effect sizes alongside p-values.
Results
This section in well-structured.
Use consistent P-value formatting and add 95% confidence intervals where possible. Consider reporting effect sizes (e.g., odds ratios) to complement prevalence data.
Discussion
The discussion could more explicitly state what new knowledge this study contributes.
Clarify why this is the first study to directly quantify false-negative rates for cartilage defects in small finger joints using histological methods.
About clinical utility of these findings: Does this suggest that radiographic OA diagnosis essentially confirms irreversible cartilage loss? What are the implications for early screening?
How should clinicians interpret normal X-rays in symptomatic patients? Could MRI or ultrasound play a complementary role?
Since the study is cadaver-based, emphasize that findings may not directly reflect disease progression in younger or symptomatic populations.
Conclusion
The conclusion section repeats the discussion findings, please avoid redundancy.
References Appropriate
Tables and figures appropriate.
Author Response
We would like to thank you for your valuable advices. We have revised the manuscript to comply with your advice as much as possible. The revised parts are marked in yellow. Please check them. We hope that our research will reach wide readers.
Comments1: Please restructure following a logical flow:
Burden and clinical importance of OA
Focus on hand OA
Pathological understanding and histological assessments
Limitations of current diagnostic tools
Study aim.
Response1: Thank you for your advices. Regarding introduction part, we have divided the paragraphs and reworded the sentences to make the logical flow easier to see.
Comments2: There is no mention of ethical approval or institutional oversight for cadaveric research.
Response2: We have added information regarding ethics committee approval and consent from the cadaver donors to the Materials and Methods section.
Comments3: Expand details on imaging parameters, fixation/storage, and histology methods for reproducibility.
Response3: We have added information about "Image Parameters" and "fixation/storage/histology methods" to the Materials and Methods section.
Comments4: Provide justification for statistical choices and, if feasible, report effect sizes alongside p-values.
Response4: We have added information about "justification for statistical choices" to the Materials and Methods section.
Comments5: report effect sizes alongside p-values.
Response5: We have added information about "effect sizes" to the Results section.
Comments6: Use consistent P-value formatting and add 95% confidence intervals where possible.
Response6: We have standardized the format of "p-value and 99% CI " in the Results section.
Comments7: Consider reporting effect sizes (e.g., odds ratios) to complement prevalence data.
Response7: We have added information about "odds ratios " to the Results section.
Comments8: Clarify why this is the first study to directly quantify false-negative rates for cartilage defects in small finger joints using histological methods.
Response8: As noted at the end of the Discussion section 4.1., “This study clarifies the false-negative rate in detecting cartilage defects by X-ray images and provides new insights into the relationship between joint evaluation using X-rays images and changes in cartilage within the joint.”
However, it is difficult to answer the question, "Why is this the first report of its kind?" To our knowledge, there have been no similar studies involving a large sample size.
Comments9: About clinical utility of these findings: Does this suggest that radiographic OA diagnosis essentially confirms irreversible cartilage loss?
Response9: Yes, as you pointed out, based on our results (PPV 100% in the results section), we believe that a radiographic diagnosis of OA strongly suggests essentially irreversible cartilage loss.
Comments10: What are the implications for early screening?
How should clinicians interpret normal X-rays in symptomatic patients? Could MRI or ultrasound play a complementary role?
Response10: Yes, clinically, for patients with symptomatic OA who show no radiographic evidence of OA, it is advisable not only to not rule out the possibility of OA but also to perform additional, more sensitive tests. MRI and ultrasound are considered strong candidates. Criteria for clinically applicable complementary imaging evaluations include the ability to observe soft tissues around the joint with high resolution, minimal invasiveness, and relatively low cost. We have added the needs for additional testing to the Clinical Relevance section. Thank you for your helpful advices.
Comments11: Since the study is cadaver-based, emphasize that findings may not directly reflect disease progression in younger or symptomatic populations.
Response11: Agree. We have added the following to the limitations: " Therefore, the results of this study do not directly reflect the conditions of groups with different backgrounds, such as populations experiencing symptoms like pain, groups of different age ranges, or other ethnic groups."
Comments12: The conclusion section repeats the discussion findings, please avoid redundancy.
Response12: The conclusion paragraph has been rewritten to avoid repeating the results and discussion.
Reviewer 3 Report
Comments and Suggestions for Authors
Thank you for the opportunity to review this clinically relevant cadaveric study. The work addresses an important gap, but several clarifications would strengthen it: please define the macroscopic “cartilage defect” criteria precisely, clarify the radiograph-reading workflow and how disagreements were resolved, and re-analyze outcomes with clustering (e.g., mixed-effects or GEE) given multiple joints per hand. Present standard diagnostic metrics (sensitivity, specificity, PPV, NPV) with 95% CIs and consider expanding histologic sampling or tempering conclusions accordingly. Improving figure/table labeling (denominators, scale bars, uniform abbreviations), tidying references, and a light language edit would further enhance readability. I believe these revisions will substantially improve the manuscript’s rigor and impact.
Comments on the Quality of English Language
The English could be improved to more clearly express the research.
Author Response
We would like to thank you for your valuable advices. We have revised the manuscript to comply with your advice as much as possible. The revised parts are marked in yellow. Please check them. We hope that our research will reach wide readers.
Comments1: please define the macroscopic “cartilage defect” criteria precisely,
Response1: Thank you for your practical and valuable advices.
We have added the criteria for macroscopic cartilage defects to the Materials and Methods section. “The joint surface where cartilage is preserved and the joint surface where the cortical bone is exposed due to cartilage loss can be easily distinguished by differences in color and texture.”
Comments2: clarify the radiograph-reading workflow and how disagreements were resolved,
Response2: Agree. To provide additional information about the radiograph-reading workflow of this study, the following has been added to the Materials and Methods section. “The radiologist's evaluation was used for analysis, and the hand surgeon's evaluation was used only to calculate the inter-rater agreement.”
Comments3: and re-analyze outcomes with clustering (e.g., mixed-effects or GEE) given multiple joints per hand.
Response3: Clustering analysis of OA distribution was not performed in this study because it was not the main purpose of this study.
Comments4: Present standard diagnostic metrics (sensitivity, specificity, PPV, NPV) with 95% CIs
Response4: The 95% confidence intervals were added for the sensitivity, specificity, PPV, and NPV of radiographic cartilage defects.
Comments5: and consider expanding histologic sampling or tempering conclusions accordingly.
Response5: All of the cadavers used in this study have already been cremated and handed over to their families, so further histological examination is not possible.
Comments6: Improving figure/table labeling (denominators, scale bars, uniform abbreviations),
Response6: I have revised some of the labels in the diagrams following your advice.
Comments7: tidying references,
Response7: References have been added.